# Unveiling Chemical, Antioxidant and Antibacterial Properties of *Fagonia indica* Grown in the Hail Mountains, Saudi Arabia

**DOI:** 10.3390/plants12061354

**Published:** 2023-03-17

**Authors:** Abdel Moneim E. Sulieman, Eida Alanaizy, Naimah A. Alanaizy, Emad M. Abdallah, Hajo Idriss, Zakaria A. Salih, Nasir A. Ibrahim, Nahid Abdelraheem Ali, Salwa E. Ibrahim, Bothaina S. Abd El Hakeem

**Affiliations:** 1Department of Biology, College of Science, Hail University, Hail 2440, Saudi Arabia; doodi23@outlook.sa (E.A.); n-almansor@hotmail.com (N.A.A.); 2Department of Science Laboratories, College of Science and Arts, Qassim University, Ar Rass 51921, Saudi Arabia; emad100sdl@yahoo.com; 3Department of Physics, College of Science, Imam Mohammad Ibn Saud Islamic University (IMSIU), Riyadh 11623, Saudi Arabia; hiidriss@imamu.edu.sa; 4Deanship of Scientific Research, Imam Mohammad Ibn Saud Islamic University (IMSIU), Riyadh 11432, Saudi Arabia; 5Department of Research and Training, Research and Training Station, King Faisal University, Al-Ahsa 31982, Saudi Arabia; zakariasalih@yahoo.com; 6Department of Biology, Faculty of Science, Imam Mohammad Ibn Saud Islamic University (IMSIU), Riyadh 11432, Saudi Arabia; nasir.adam@yahoo.com; 7Department of Home Economic, College of Home Economic, King Khalid University (KKU), Guraiger, Abha 62529, Saudi Arabia; Nahid.Nemir@hotmail.com (N.A.A.); salamin@kku.edu.sa (S.E.I.); bothaina_2007@yahoo.com (B.S.A.E.H.)

**Keywords:** bioactive substances, microorganisms, disc-diffusion, phenol, tannins, alkaloids, saponins, minerals, anti-biofilm

## Abstract

The Aja and Salma mountains in the Hail region are home to a variety of indigenous wild plants, some of which are used in Bedouin folk medicine to treat various ailments. The purpose of the current study was to unveil the chemical, antioxidant and antibacterial properties of *Fagonia indica* (Showeka) grown widely in these mountains, as data on the biological activities of this plant in this remote area are scarce. XRF spectrometry indicated the presence of some essential elements, which were in the order of Ca > S > K > AL > CL > Si > P > Fe > Mg > Na > Ti > Sr > Zn > Mn. Qualitative chemical screening revealed the presence of saponins, terpenes, flavonoids, tannins, phenols and cardiac glycosides in the methanolic extract (80% *v*/*v*). GC–MS showed the presence of 2-chloropropanoic acid 18.5%, tetrahydro-2-methylfuran 20.1%, tridecanoic acid 12-methyl-, methyl ester 2.2%, hexadecanoic acid, methyl ester 8.6%, methyl 3-(3,5-di-tert-butyl-4-hydroxyphenyl) propionate 13.4%, methyl linoleate 7.0%, petroselinic acid methyl ester 15%, erucylamide 6.7% and diosgenin 8.5%. Total phenols, total tannins, flavonoids, DPPH, reducing power, -carotene and ABTS IC_50_ (mg/mL) scavenging activity were used to measure the antioxidant capabilities of *Fagonia indica*, which exhibited prominent antioxidant properties at low concentrations when compared to ascorbic acid, butylate hydroxytoluene and beta-carotene. The antibacterial investigation revealed significant inhibitory effects against *Bacillus subtilis* MTCC121 and *Pseudomona aeruginosa* MTCC 741 with inhibition zones of 15.00 ± 1.5 and 12.0 ± 1.0 mm, respectively. The MIC (minimum inhibitory concentration) and MBC (minimum bactericidal concentration) ranged between 125 to 500 μg/mL. The MBC/MIC ratio indicated possible bactericidal efficacy against *Bacillus subtilis* and bacteriostatic activity against *Pseudomona aeruginosa*. The study also showed that this plant has anti-biofilm formation activity.

## 1. Introduction

Medicinal plants have been used as drugs since ancient times. The interest in medicinal plants has resurged in recent decades due to tremendous scientific reports that showed their remarkable in vitro efficacy as antimicrobial agents [1], antiviral agents [2], antioxidants [3], anticancer agents [4], and many more bioactive properties. In the current century, with the development of medical research, the severity of illness has also risen. For the treatment of complex disorders, new medications are being created, but these medications also come with a variety of side effects, ranging in severity from mild to severe. According to the World Health Organization (WHO), multidrug-resistant tuberculosis and Gram-negative bacteria have become a global health-threatening issue, and the WHO has called on member states to make greater efforts into researching and developing new antibiotics to control this dilemma [5]. The use of herbs for their therapeutic benefits is known as herbal medicine, sometimes known as herbalism or botanical medicine [6]. Numerous studies have shown that certain plants contain various compounds, such as peptides, aldehydes, alkaloids, phenols, saponins, polysaccharides, terpenoids, steroids, flavonoids and tannins, which may have critical restorative effects against bacteria, fungi, or viruses [7,8,9,10]. Herbal plants produce and contain a variety of chemical substances that have an impact on humans. Many higher plants have been used as anti-infection remedies [11].

On the other hand, it seems that natural remedies are more potent than their synthetic counterparts, and have minimal side effects (Nisar et al., 2018). Oxidative stress is a good example for this claim. Oxidative stress is a state in which oxidative powers are stronger than antioxidant systems because the balance between them has been upset [12]. Plants produce a large amount of antioxidants, such as phenolic compounds, flavonoids, polyphenolics, and tannins, that stop the damage that reactive oxygen species can do to cells; therefore, medicinal plants have a lot of antioxidants that are beneficial for health maintenance [13].

The Arabian Peninsula is desert-dominated and biodiversity-poor. However, the Kingdom of Saudi Arabia (KSA) boasts a diverse flora, including many edible and medicinal plants and many types of trees, herbs, and shrubs, although scientists have not yet shown significant interest in its potential therapeutic compounds [14]. The Kingdom of Saudi Arabia’s natural plant life inhabits a considerable portion of the Hail area, which is also home to many animals and plants. Numerous plant species from various families and orders have been found in the Hail area. Native plants in Hail were frequently used for multiple purposes, such as food, herbal medicine and grazing animals’ nutrition. The Aja and Salma Mountains (Figure 1) and Jabal Rumman are only a few of the mountains that make up the Hail region’s diverse topographical environment. In addition, there are several valleys, including Wadi Al-Adi’a and Al-Hamima, and several reefs, including Sha’ab Al-Sahba and Shu’ib Shatib. In addition, the area is home to plains, dunes, streams and several small pools of water that emerge after rain [15,16].

*Fagonia* spp. is a desert plant widely spread throughout the Middle east, North Africa, the Horn of Africa, Iran, and India. It is a woody tree or shrub that can grow to a height of two, three, and occasionally five meters. The plant is grayish-green in color, and its branches are basal. A white cork husk covers the stem, and when the husk is scraped, a viscous milky liquid leaks out. Taxonomically, *Fagonia* is a member of the *Zygophyllaceae* family, which includes over 22 genera and more than 250 types. Regarding the morphology, *Fagonia indica* is considered one of the most important members of the genus *Fagonia*. It is a tiny, spiky bush that grows to a height of 60 cm and a width of about 100 cm [17,18]. It has been used as a medicinal plant in folk medicine in Pakistan, India, and the Far East, and it is rumored to have unique therapeutic benefits and tremendous goodness. It has a wide range of traditional applications, including for diarrhea, crises, urinary secretions, and liver issues. Additionally, they may have less effectiveness as a powerful antibacterial agent against various hazardous microorganisms, due to neck abscesses. Research has demonstrated that the plant affects blood pressure, the neurological system, cancer, and liver disorders [19,20]. The effectiveness of these mixes against bacteria needs to be better understood, and scant information is available on the antibacterial characteristics, as well as on the chemical and antioxidant characteristics of the plant under study from this remote area. To the best of our knowledge, there are no data on the biological activities of *Fagonia indica* grown in the Hail Mountains, Saudi Arabia. Therefore, the current study aimed to investigate the chemical composition of *Fagonia indica*, and to determine its possible antibacterial and antioxidant activities using different standardized methodologies.

## 2. Results and Discussion

### 2.1. Phytochemical Screening of Fagonia indica

Most people concur that plants’ medicinal properties are due to their bioactive phytochemical components [21]. Thus, investigations on plant components can provide scientific justification for the traditional uses of these plants. The results of the phytochemical analysis revealed the presence of cardiac glycosides, saponins, terpenes, flavonoids, tannins, and phenols in the methanol extract (80% *v*/*v*) from *Fagonia indica* (Table 1). Phytochemical compounds are secondary metabolites generated by plants for specific activities essential to the plant’s survival in its environment, as well as its for capability to withstand biotic and abiotic stresses. These substances have direct or indirect multi-functional effects on eukaryotic cells, prokaryotic cells and viruses, but have no function in plant growth or development [22]. As an example, inflammation is hypothesized to be reduced by polyphenols and flavonoids by inhibiting the production of pro-inflammatory chemicals [23]. Additionally, the polyphenols are antimicrobial. Since many pathogenic microorganisms have developed the ability to survive current treatments at clinically significant concentrations, their eradication may benefit from antibacterial action [24].

### 2.2. GC–MS Analysis of Fagonia indica

Gas chromatography coupled with mass spectroscopy is a crucial tool for analyzing chemical substances. The mass spectrum supplies qualitative data about a substance’s chemical constituents. The GC–MS chromatogram of the methanol extract of *Fagonia indica* is shown in (Figure 2). The peak retention time, peak area percentage, height percentage and mass spectral fragmentation features of the chromatograms were compared with those of recognized compounds in the library of the National Institute of Standards and Technology (NIST) [25]. From the GS–MS chromatogram, nine peaks were identified. As presented in Table 2 and Figure 3 are the chemical compounds and their percentages in methanolic extract of *Fagonia indica*. The detected phytochemicals were 2-chloropropanoic acid 18.5%, tetrahydro-2-methylfuran 20.1%, tridecanoic acid, 12-methyl-, methyl ester, 2.2%, hexadecanoic acid, methyl ester 8.6%, methyl 3-(3,5-di-tert-butyl-4-hydroxyphenyl) propionate 13.4%, methyl linoleate 7.0%, petroselinic acid methyl ester 15%, erucylamide 6.7% and diosgenin 8.5%. According to the findings, tetrahydro-2-methylfuran accounted the highest percentage of phytochemicals (20.1%), whereas tridecanoic acid, 12-methyl-, methyl ester accounted the lowest percentage (2.2%) of the chemicals, as depicted in Figure 2. The compounds identified were primarily plant fatty acids (2-chloropropanoic acid, tridecanoic acid, 12-methyl-, methyl ester, hexadecanoic acid, methyl ester, methyl 3-(3,5-di-tert-butyl-4-hydroxyphenyl) propionate, methyl linoleate, petroselinic acid methyl ester and erucylamide), carbohydrates (tetrahydro-2-methylfuran) and sterols (diosgenin). It has been documented the literature that *Fagonia indica* contains a number of phytochemicals, including flavonoids, alkaloids, terpenoids, saponins, tannins, sterols, anthraquinones and coumarin. However, phytochemicals, such as flavonoids, alkaloids, terpenoids, tannins, coumarin and anthraquinones, were not detected in the current study. Meanwhile, the presence of alcohols, fatty acids, heterocyclics and esters have been reported in methanolic extracts of this plant [26]. On the other hand, a GC–MS analysis of many non-polar and polar plant extracts demonstrated the existence of flavonoids, terpenoids, alkaloids, tannins and sterols [27,28]. It is well documented the majority of sterol compounds have anti-cancer, anti-inflammatory, immunomodulatory, and even antiviral properties [29]. Although their therapeutic effects have not been established, carbohydrates may enhance the efficacy of other physiologically important substances. Furthermore, carbohydrates have also been employed to create polysaccharide immunomodulatory effects, which have potential uses in vaccines and in medicine. As a result, the combined active ingredients found in each plant may have greater therapeutic efficacy than a single isolated molecule [29]. Numerous biological processes involving fatty acids, such as their anti-inflammatory characteristics, help organisms defend themselves [30]. According to the findings of a number of studies, *Fagonia indica* can both aid in the prevention of bacterial and fungal diseases, and speed up the body’s recovery from their effects [31].

### 2.3. Energy Dispersive X-ray Fluorescence Analysis

The energy dispersive X-ray fluorescence spectrometer is a non-destructive detector that has the ability to detect a wide range of elements, from Na to Am. The elemental analysis of the *Fagonia indica* plant resulted in the detection of 14 elements. Table 3 presents the concentrations of K, Ca, Ti, S, Ni, Mn, Fe, AL, Cl, Si, Na, Sr, Zn and Mg detected in a *Fagonia indica* sample. From the table, it can be noticed that the elemental concentrations were in the order of Ca > S > K > Al > Cl > Si > P > Fe > Mg > Na > Ti > Sr > Zn > Mn. The accumulated metal ions had both direct and indirect effects on bacteria. By integrating into bacterial proteins, the dissolved ions had an indirect impact on bacteria by rendering those proteins inactive or dysfunctional. It is believed that when metals dissolve, active radicals are created that directly harm bacteria, causing the cell wall to burst and leading to death [32]. Furthermore, using energy dispersive X-ray fluorescence examination, Table 3 demonstrates that no heavy metals such as Pb and Cd were found. In recent years, the presence of some heavy metals such as Cd, Co, Cr and Cu, due to the accumulation of residues and pollutants in agricultural soil and in cultivated medicinal plants, has become an issue of interest and research, as medicinal plants are freely sold in open-air markets without chemical or biological analysis [33,34].

### 2.4. Antioxidant Activities

Free radicals are a byproduct of the body’s usual metabolic activity in biological systems. By scavenging free radicals, antioxidants defend us against several ailments [35]. Total phenols, total tannins, flavonoids, DPPH, reducing power, beta-carotene, and ABTS IC_50_ (mg/mL) scavenging activity were used to measure the antioxidant capabilities of *Fagonia indica* in comparison to unveiled standard molecules (Table 4). The acquired results show promising antioxidant properties at low concentrations when compared to ascorbic acid (AA), butylate hydroxytoluene (BHT) and beta-carotene, as compared with the results of [35,36].

Total phenols, total tannins, flavonoids, DPPH, reducing power, -carotene and ABTS IC_50_ (mg/mL) scavenging activity were used to measure the antioxidant capabilities of *Fagonia indica* in comparison to well-known standard molecules (Table 3). Comparing the obtained values to those for ascorbic acid (AA), butylate hydroxytoluene (BHT) and β-carotene, the data show potential antioxidant benefits at low doses.

*Fagonia indica* contained the following amounts of total phenols, total tannins and total flavonoids: 3.72 ± 0.05 mg GAE/g, 24.14 ± 0.65 mg TAE/g and 136.05 ± 0.45 mg QE/g, respectively. The addition of an antioxidant can drastically reduce the stability of the free radical DPPH. In this research, it was investigated at what sample concentrations the DPPH radical scavenging activity was 50% inhibited (IC_50_). The methanol extract of *Fagonia indica* had an average IC_50_ value of 00.06 ± 3.53 (mg/mL), which suggests significant antioxidant activity.

The presence of an antioxidant can quickly reduce the stability of the free radical DPPH, and it absorbs (UV-Vis) strongly and visibly at 517 nm. The methanol solutions’ decreased ability to absorb the DPPH radical at 517 nm indicates increased antioxidant activity. In this experiment, the sample concentrations that resulted in a 50% inhibition (IC_50_) of the DPPH radical scavenging activity were found. DPPH-scavenging action and efficiency of the extraction solvents was compared in this research using the IC_50_ value of the DPPH scavenging activity. A lower IC_50_ value indicated more antioxidant activity.

The results of this study revealed that the plant contains tannins, flavonoids and total phenolic content. The different extracts can bleach stable ABTS and O_2_ free radicals; researchers can assess their capacity for free radical scavenging. The extracts’ capacities to neutralize O_2_ and ABTS radicals were significantly lower than those of the standards.

Polyphenols (such as phenolic acids, tannins, coumarins, anthraquinones and flavonoids) with redox properties—enabling them to function as reducing agents, singlet oxygen quenchers, hydrogen donors, and have the potential to chelate metal ions—make up the majority of the antioxidants isolated from higher plants. Through the use of the reagent Folin–Ciocalteu phenol, the total polyphenol content of *Fagonia indica* was determined to be 3.72 ± 0.05 mg gallic acid equivalent dry weight of plant per gram.

### 2.5. Antibacterial Activity

Results of the antibacterial activity of *Fagonia indica* methanol extract are shown in (Table 5). The plant extract successfully suppressed the studied microorganisms, which represented Gram-negative bacteria (*Pseudomonas aeruginosa* MTCC741) and Gram-positive bacteria (*Bacillus subtilis* MTCC121), at different concentrations (1, 2, and 3 mg/disc); the last concentration (3 mg/disc) was significant compared to the reference drug (ampicillin 10 μg/disc) at *p* ≤ 0.05. At 3 mg/disc of the methanol extract of *Fagonia indica*, *Bacillus subtilis* recorded 15.00 ± 1.5 mm and *Pseudomonas aeruginosa* recorded 12.0 ± 1.0 mm inhibition zones. The Gram-positive bacteria exhibited the highest zone of inhibition. This may be explained by the fact that, in contrast to Gram-positive bacteria, the lipopolysaccharide layer and periplasmic zone around Gram-negative bacteria shield the cell membrane from the damaging effects of the plant extract [37]. The MIC and MBC values confirmed the findings of the disc diffusion test (Table 6). Bacillus subtilis was found to be the most susceptible bacterium (MIC = 125 μg/mL, MBC = 250 μg/mL), while Pseudomonas aeruginosa was found to be the least susceptible (MIC = 250 μg/mL, MBC = 500 μg/mL). To understand the mechanism of the antibacterial activity, the MBC/MIC ratio was calculated, and it was found to be 4 for *Bacillus subtilis* and 16 for *Pseudomonas aeruginosa*. In general, for any tested extract, if MBC/MIC is less than or equal to 4, it is considered bactericidal, and if it is higher than 4, it is considered bacteriostatic [38]. Our study is in agreement with some previous studies; worldwide, it was cited that *Fagoina indica* collected from the Cholistan desert in India showed substantial antibacterial activity against *Escherichia coli* [39]. *Fagoina indica* grown in the desert near Dubai, United Arab Emirates, exhibited good antimicrobial activity against a group of Gram-positive and Gram-negative bacteria, and also showed anti-Candida activity at a concentration of 200 mg/mL [40]. In fact, the first stage in creating new chemotherapeutic medications from plants, which are a substantial source of potentially beneficial compounds, is to assess the in vitro antimicrobial activity. Many researchers have examined plants’ antibacterial and antifungal properties [41,42]. Some of these discoveries have aided in identifying the underlying reason for specific behaviors, leading to the development of drugs that may be used to treat them and enhance people’s well-being. Plants with potential antibacterial effects should be evaluated against a suitable microbiological model to establish the activity and identify its associated features.

The alcoholic and aqueous extracts of numerous plants and herbs effectively suppress food pathogens and spoilage bacteria with various potencies. Few reports exist regarding the effectiveness of the *Fagonia indica* plants’ antibacterial compounds against harmful pathogens. Only a few substances—many of which have antimicrobial effects—can be regarded as therapeutic, because mammalian cells are more sensitive to chemical inhibition than microbial cells. In the current study, we investigated the antibacterial activity of *Fagonia indica* methanolic extract against the pathogenic bacteria *B. subtilis* and *P. aeruginiosa*. It was revealed from earlier research that *Fagonia cretica* methanol extract showed high antibacterial activity [43,44]. The antibacterial activity of the *Fagonia indica* extract may be explained by the presence of several bioactive components, such as saponins, flavonoids and alkaloids, found in the methanolic extract as well as in the aqueous extract of *Fagonia indica* (Table 1 and Table 2). These results were in agreement with those of Anil et al. [27], who suggested that the diverse antibacterial activity of the plant extract is explained by the presence of several bioactive components, such as saponins, flavonoids and alkaloids. Previous research found that the alcoholic extract of many plants such as *Piper stylosum*, *Epipremnum* sp., *Tetracera indica*, *Zingiber* sp., *Tectaria crenata*, *Goniothalamus* sp., *Homalomena propinque*, *Smilax* sp., *Elephantopus scaber*, *Mapania patiolale*, *Melastoma* sp., *Phullagathis rotundifolia*, *Stemona tuberosa* and *Thotea grandifolia* could inhibit the growth of *Bacilus subtilis* [45].

The results of the anti-biofilm formation of *Fagonia indica* against *B. subtilis* and *P. aeruginosa* are shown in Table 7. This plant exhibited noticeable anti-biofilm formation activity. The extract of *Fagonia indica* had an MIC value of 125 mg/mL and an MBC value of 500 mg/mL, according to the preliminary antibacterial investigation. The inhibition of biofilm by these extracts ranged from 40.19 to 20.60% for *B. subtilis*, and from 33.69 to 17.78% for *P. aeruginosa*. Different levels of anti-biofilm activity against the examined bacteria were present in the plant species of this study. However, we contend that *Fagonia indica* is a strong contender, and more research is required to identify the antimicrobial substances for the control of multidrug-resistant pathogenic bacteria and their modes of action. Biofilms are multi-species bacterial communities with intricate structures that lead to antibiotic resistance and potentially fatal illnesses, resulting in significant financial loss; new methods are required. As a result of their therapeutic and antibacterial properties, medicinal herbs are being studied as potential replacements [46].

There are currently no “anti-biofilm drugs” that have been approved for use in people, even though microbial biofilms have been linked for two to three decades to antibiotic resistance and persistent microbial infections. To treat infectious disorders linked to biofilms, it is imperative to create new “anti-biofilm” drugs. Numerous studies indicate that blocking molecules involved in quorum sensing or biofilm-specific transcription may prevent the growth of biofilms. The idea of focusing on other important microbial biofilm components, notably the extracellular matrix components, has received very little attention [47,48].

Verma et al. [49] reported that two small molecule inhibitors (lovastatin and simvastatin), discovered by virtual screening and pharmaceutical repurposing, completely inhibited biofilm. These inhibitors target the significant proteinaceous portion of *B. subtilis*. Another result of these putative inhibitors was the breakdown of already-formed biofilms, which suggested that innovative anti-biofilm therapy strategies against biofilm-forming chronic microbial diseases could be developed using a similar method that uses FDA-approved medications to target ECM-associated proteins.

*P. aeruginosa* is a widespread opportunistic infection that has been connected to severe morbidity and mortality in some groups. Environments containing soil, water and hosts, such as plants, animals and people, are ideal for it to flourish [50]. It is generally present in high amounts in everyday meals, particularly vegetables. Additionally, it is present in trace amounts in drinking water. Its adaptable energy metabolism would be the reason for its universality. *P. aeruginosa* develops biofilms to colonize various surfaces, just like other bacterial species that are common in the environment (including food packaging, water taps and medical devices). As a result of this, the cells resist both host defenses mediated by macrophages, neutrophils and antibacterial agents, including antibacterial cleansers, disinfectants and therapeutically relevant antibiotics. Several studies have successfully employed plant extracts to manage *P. aeruginosa* in the recent past. To the best of our knowledge, this is the first study on the antibacterial characteristics of *Fagonia indica* grown in mountains in the Hail region, Saudi Arabia.

## 3. Materials and Methods

### 3.1. Sample Collection

*Fagonia indica* (Showeka) is grown in the Hail region, situated in the northern middle part of Saudi Arabia 25 290 N and 38 420 E. (Figure 4). It occupies 118,322 square kilometers, or 6% of the total land of Saudi Arabia. *Fogonia indica* (Showeka) samples were taken from Aja Mountain (27° 259,040 N, 41° 259,330 E) during the period between March to May, 2021.

### 3.2. Chemicals

All of the chemicals (methanol, ethanol, chloroform) used were of analytical grade. The chemicals and indicators were utilized as received, without any purification, and were obtained from Merck (Merck KgaA, Darmstadt, Germany). Folin–Ciocalteu’s phenol reagent, aluminium chloride hexahydrate, quercetin, H_2_SO_4_, standard vitamin E and DPPH (2,2-Diphenyl-1-picrylhydrazyl) reagents were from Sigma-Aldrich (St. Louis, MI, USA).

### 3.3. Preparation of Plant Extract

Fresh samples of *Fagonia indica* areal parts (whole plant) were gathered and identified by the Department of Biology, College of Science, University of Hail, Saudi Arabia (Figure 5). The obtained *Fagonia indica* was dried in a shad for up to a week. Then, whole plant material was ground using a mechanical grinder (Electric Grinder, OMCG2145, Olsenmark, New York, NY, USA) to obtain a smaller particle size that was preferable for solvent extraction efficiency. In total, 100 g of the plant’s powder was macerated in 1000 mL of 80% methanol for up to three days at room temperature (about 35–37 °C), with frequent shaking. The methanol was evaporated, and the extract was concentrated using a rotary vacuum evaporator (8 kw 50 L, Henan Lanphan Industry Co., Ltd, Zhengzhou, China). Furthermore, the unfiltered extracts were injected into the GC–MS after being dissolved in hexane and microfiltered.

### 3.4. Gas Chromatography–Mass Spectrometry (GC–MS) Analysis

The GC–MS investigation used a Perkin Elmer Clarus 600 GC System equipped with an Rtx 5MS capillary column (30 m 0.25 mm i.d. 0.25 m film thickness; maximum temperature, 350 °C) and coupled to a Perkin Elmer Clarus 600C MS. As the carrier gas, a steady flow of 1.0 mL/min of ultra-high-purity helium (99.9999%) was employed. The ion source, transfer line and injector temperatures were 280 °C, 270 °C and 270 °C, respectively. The gas had an ionization energy of 70 eV. The electron multiplier (EM) voltage was derived from auto tune. All data were collected using full-scan mass spectra between 40 and 550 amu [29,51].

### 3.5. GC–MS Analysis Conditions

The split ratio of the 1 L injected sample was 10:1. The oven’s temperature program was set to hold at 280 °C for 25 min, at a rate of 80 °C per minute from 60 °C. The entire performance lasted 53.5 min. Conditions for the GC–MS analysis of the leaf oil: the GC–MS apparatus, as previously noted, detected fatty acid methyl ester (FAME) molecules. The flow of helium gas was 0.7 mL/min. The temperatures of the ion source, transfer line and injector were 250 °C, 250 °C and 220 °C, respectively. Initially set at 50 °C (held for 8 min), the oven temperature was raised to 250 °C at a pace of 40 °C per minute. By gathering full-scan mass spectra throughout the scan range of 35–500 amu, all data were obtained. Through comparison of the collected spectra with mass spectral libraries, the unidentified chemicals were found [52]. The determination of the calibration and minimum detection limits for the system was based on the manufacturing conditions, and the corresponding equation is available elsewhere [53].

### 3.6. Phytochemical Profile

The extracts were tested for the determination of major phytochemical compounds using qualitative methods described elsewhere [54,55], as follows:

Detection of saponins:

The methanol plant extract (0.5 g), which was mixed in a test tube, was tested for the presence of saponins if persistent foaming was seen after warming.

Detection of terpenes:

To detect the presence of terpenes (terpenoids), a dried extract weighing 50 mg was soaked in 5 mL of ethanol and mixed with 2 mL of chloroform. The resulting mixture was slightly warmed and then cooled. To the cooled mixture, 3 mL of concentrated H_2_SO_4_ was gradually added along the sides of the test tubes. A brownish-red precipitate was formed at the interface, which confirmed the presence of terpenes.

Detection of flavonoids:

To determine the presence of flavonoids, a dried extract weighing 0.30 g was extracted with 30 mL of distilled water for 2 h, and filtered using Whatman filter paper number 42 (125 mm). Next, 10 mL of the aqueous filtrate of the extract was mixed with 5 mL of 1.0 M dilute ammonia solution, followed by the addition of 5 mL of concentrated tetraoxosulphate (VI) acid. The appearance of a yellow coloration that disappeared upon standing indicated the presence of flavonoids.

Detection of phenols:

To identify the presence of phenols, the ferric chloride test was conducted. Specifically, a 10 mL extract solution was mixed with a few drops of ferric chloride solution. The formation of a bluish-black coloration was an indication of the presence of phenols.

Detection of tannins:

To perform the tannin test, 1 mL of the extract was combined with water and subjected to heating on a water bath. After filtration, the resulting filtrate was subjected to the addition of ferric chloride. The development of a dark green hue was indicative of the existence of tannins.

Detection of alkaloids:

To identify the presence of alkaloids, two techniques are commonly used. The first is the Dragendorff’s test, which involves the addition of 2 mL of MeOH and 2 mL of 1% HCl to 5 mg of the extracts along the side of the test tube, followed by the addition of 500 μL of Dragendorff’s reagent to the mixture. A positive test is indicated by the formation of an orange or orange reddish-brown precipitate. The second method is the Mayer’s test, in which a drop or two of Mayer’s reagent is added to 1 mg/mL of the extract. The presence of alkaloids is confirmed by the formation of a white or creamy precipitate.

Detection of cardiac glycosides:

To detect the presence of cardiac glycosides in a 2.5 mg extract, a solution comprising 1 mL of glacial acetic acid and a few droplets of 5% ferric chloride was introduced. Subsequently, 0.5 mL of concentrated sulfuric acid was added along the side of the test tube. A positive result, indicative of the presence of cardiac glycosides, was determined by the formation of a green or blue hue.

### 3.7. Energy Dispersive X-ray Fluorescence Measurements

The electrical conductivity (EC) of the plant sample was measured at 20 °C with the aid of a conductivity meter [56]. An EDXRF (energy dispersive X-ray fluorescence) from Thermo Fisher Scientific (Waltham, MA, USA) was used to determine the concentrations of elements of the plant sample. The instrument was equipped with a state-of-the-art silicon drift detector (SDD) that eliminates spectral interference while offering a rapid response. The active area of 30 mm^2^ allows for a wide hard angle for efficient X-ray capture. The high flux rhodium anode tube was designed to enable direct X-ray tube excitation or custom excitation via different filters, enhancing elemental sensitivity. The calibration and minimum detection limits were determined following the conditions under which it was manufactured, and the equation can be found elsewhere [57].

### 3.8. Determination of Total Phenolic Content (TPC)

According to the procedure of Kumar et al. [58], using the Folin–Ciocalteu reagent, the total phenolic content (TPC) of different extracts was measured. The samples were analyzed at a concentration of 1 mg/mL. The extract was diluted ten-fold with deionized water before being put in a test tube with 0.75 mL of Folin–Ciocalteu reagent and stirred. The mixture was allowed to stand at 25 °C for 5 min. Following the addition of 0.75 mL of a saturated sodium carbonate solution, the liquid was gently mixed. Using a UV-Vis spectrophotometer, the absorbance at 725 nm was measured after 90 min at 25 °C. Gallic acid was used to develop a calibration curve. Gallic acid equivalents (GAE) in mg/mg of vegetable extract (mg of gallic acid/mg dry weight) were used to quantify the total phenolic content.

### 3.9. Determination of Total Flavonoid Content (TFC)

The extracts’ total flavonoids concentration (TFC) was determined with minor modifications [59]. A volume of 1.5 mL (1 mg/mL) of the extract was mixed with an equal amount of 2% AlC_l3_–6H_2_O. After 10 min of incubation, the mixture was vigorously agitated, and the absorbance at 367 nm was measured. Using a quercetin calibration line, units of milligrams of quercetin per gram of dry weight (mg Q.E./mg) were used to quantify the total flavonoid content. There were three tests performed on every sample.

### 3.10. Determination of Total Tannin Content (TTC)

The tannins were quantified with a colorimetric method using a modified vanillin test [60]. A volume of 1.5 mL of concentrated H_2_SO_4_ and 3 mL of a 4% methanolic vanillin solution were added to 50 mL of the extract (1 mg/mL). After allowing the mixture to stand for 15 min, the absorbance at 500 nm was calculated using methanol or water as a reference. The TTC was expressed as mg catechin/g of dry weight, or mg C.E./mg. Each sample underwent three replicate analyses.

### 3.11. DPPH Radical Scavenging

The DPPH radical scavenging method was utilized to evaluate the free radical scavenging activity of the extracts, with the standard as vitamin E [61]. The various extracts (stock solutions 20 mg/mL and 1 mg/mL) and the standard were pipetted into separate test tubes (stock solutions). In a volume of 0.5 mL, an equal volume of DPPH methanolic solution was added to each sample and the standard. Before being allowed to stand for 30 min at a temperature of 25 °C in the dark, the mixture was vigorously stirred. The absorbance of the resultant solution at 520 nm was measured with a spectrophotometer. Each measurement was made three times. In total, 0.5 mL of the DPPH solution and 0.5 mL of the methanol were combined as a control. Pure methanol was assumed to be the blank. To calculate the percentage (PI %) of free radical DPPH inhibition, the following equation was used:
PI % = 100 × (A of Control − A of Sample)/A of Control
(1)

where A of Control and A of Sample are the absorbances of the control solution and of a test sample or standard, in order.

### 3.12. ABTS Radical Scavenging Activity Assay

2,2′-azino-bis (3-ethylbenzthiazoline-6-sulphonic acid), commonly known as the ABTS cation scavenging activity test, was used to conduct the antiradical assay [61]. The ABTS radical mono-cation was formed by reacting a 7 mM ABTS solution with 2.45 mM K_2_S_2_O_8_. The combination was allowed to stand at room temperature and in the dark for 15 h. For the organic extracts, the samples were dissolved in methanol, and for the aqueous extract, the samples were dissolved in distilled water. The tocopherol (vitamin E) standard and various extract concentrations were examined, with the benchmark serving as a yardstick. By mixing 800 mL of diluted ABTS+ with 200 mL of each standard and sample, the anti-oxidant activity was determined. The absorbance was measured spectrophotometrically at 734 nm after thirty minutes. Each measurement was performed thrice. The antioxidant capacity of the test samples and the standard was represented as a percentage (%) of inhibition. The proportion of ABTS+ scavenging was computed using the following equation:
PI % = 100 × (A of Control − A of Sample)/A of Control
(2)

where A of Control and A of Sample are the absorbances of the control and of the test sample or a standard, respectively.

### 3.13. β-Carotene/Linoleic Acid Method

The extracts’ capacity to stop -carotene from bleaching was evaluated using a method previously published by [62]. Linoleic acid produces a free radical when it is heated with the compound of β-carotene and linoleic acid. Two milliliters of the β-carotene solution (1.5 mg -carotene/2.5 mL chloroform) were added to twenty milliliters of linoleic acid and two hundred milliliters of Tween-20. The chloroform was evaporated at 40 °C in a vacuum with a revolving evaporator. The dried material was mixed with 50 mL of distilled water to make a β-carotene-linoleic acid emulsion. By adding 0.800 mL of the emulsion to 0.200 mL of extracts at varied concentrations (stock solution 20 mg/mL), we examined each extract’s capacity to bleach -carotene and observed the results. The absorbance at 470 nm was measured before and after the combinations were incubated for 120 min at 50 °C in a water bath. Each test was conducted three times. The antioxidant activity of extracts was calculated using the following formula:
(3)$$\mathrm{PI}\;\%\;\mathrm{is}\;\mathrm{calculated}\;\mathrm{as}:\;1-\left(A0-At/Ac0-Act\right)\times 100$$
where *A*0 and *Ac*0 are the test sample’s, standard’s, or control’s absorbance values recorded at zero time, respectively, and *At* and *Act* are the corresponding absorbance values measured after incubation for 120 min, respectively.

### 3.14. Bacterial Strains

As a test bacterium, samples of the crude extract of the *Fagonia indica* plant were tested for their ability to inhibit the growth of *Bacillus subtilis* (MTCC121) and *Pseudomonas aeruginosa* (MTCC741). Both strains of bacteria were received from the Microbial Type Culture Collection (MTCC), Chandigarh, India, and then cultivated on Muller–Hinton agar. A single colony of bacteria was transferred to fresh media and left to rise overnight at 37 °C to form bacterial cultures. Using the sterile saline solution, the culture’s turbidity was adjusted to match the 0.5 McFarland standard (10^6^ CFU/mL).

### 3.15. Agar Disc-Diffusion Method

The disc-diffusion test was used to evaluate *Fagonia indica*’s antibacterial activity. *Bacillus subtilis* MTCC121 and *Pseudomonas aeruginosa* MTCC741 suspensions of the studied microorganisms were made, adjusted to McFarland turbidity to obtain about 10^6^ CFU/mL, and then swapped over to sterile plates containing 20 mL of Mueller–Hinton agar. Individual sterile filter discs, 5 mm in diameter (Whatman No. 1), were put into previously infected agar plates under aseptic conditions after being saturated with 1, 2 and 3 mg/disc of the methanolic extract. At room temperature, the plates were incubated for 24 h. Discs impregnated solely with 80% methanol were used as a negative control. Ampicillin, at 10 μg/disc and as a positive control disc, served as the standard drug. The mean value was calculated from three individual repetitions [63].

### 3.16. Determination of Minimum Inhibitory Concentration (MIC) and Minimum Bactericidal Concentration (MBC)

The minimum inhibitory concentration (MIC) of *Fagonia indica* methanolic extract of was estimated using a microdilution method on 96-well microplates [64]. In a nutshell, decreasing concentrations of the extract (500 to 31.25 μg/mL) were produced in 5% DMSO for each microplate row using the serial two-fold dilution method. Following that, the microplates were incubated with 20 μL of bacterial suspensions adjusted to 0.5 McFarland and 160 μL of Mueller–Hinton broth for 24 h at 37 °C. The bacterial growth was then tested by incubating 40 μL of 2, 3, 5-triphenyltetrazolium chloride (TTC) (at a concentration of 0.2 g/mL for 30 min at 37 °C. The TTC identifies the wells with bacterial growth by staining the bacterial cells with a red dye. Microplate wells with the least amount of extract and no discernible bacterial growth were taken as the MIC. The minimum bactericidal concentration (MBC) of the *Fagonia indica* methanol extract against the examined bacteria was carried out following the diffusion test described in the literature [65], with minor modifications. The Mueller–Hinton agar plates were loaded with 50 μL of each of the tubes from the MIC test that exhibited no apparent growth with *B. subtilis* and *P. aeruginosa*, and these plates were subsequently incubated for up to 24 h at 35–37 °C. The growth of bacteria on the plates was checked after the incubation time. The lowest concentration that did not allow even a single bacterial colony to grow on the Mueller–Hinton agar plate was defined as the MBC value. The MBC/MIC ratio was also calculated.

### 3.17. Anti-Biofilm Assays

The biofilms of each bacterial strain were produced on 96-well microtiter plates with MHB, 1% glucose and cells (108 cells/mL) for 24 h at 37 °C [66]. Planktonic cells were delicately removed after incubation, and the wells underwent three N-saline washes. Following that, 200 mL of crude extract (Sub-MIC) was added to the wells, which were then incubated for 24 h at 37 °C. At 0 and 24 h, the absorbance at 492 nm was measured. The positive control used was chloramphenicol. All of the assays were run in triplicate. With the aid of M.H.B. medium containing distinct bacterial strains, biofilm formation was inhibited. The amount of biofilm inhibition was calculated as follows:**OD (control) − OD (test)/OD (control) × 100**(4)

### 3.18. Statistical Analysis

The data were analyzed using the SPSS application for Windows, version 20 (SPSS Inc., Chicago, IL, USA). Based on Duncan’s multiple-range test, a one-way ANOVA was used to determine the difference between the groups. The information was displayed as means and S.D., and statistics were judged to be significant at *p ≥ 0.05*.

## 4. Conclusions

The findings of this study on the phenolic content, antioxidant properties and antibacterial capabilities of the methanolic extract of *Fagonia indica* validated the plant’s folkloric use as medicine by Bedouin in the Hail region, and established a framework for further research into its potential as a functional food. Future research on the connection between structure and bioactivity will require purifying and separating the extract’s numerous components. Based on conventional knowledge, responsible phytoconstituents and targeted compounds were isolated and identified. Strong scientific data suggests that this plant should be studied in therapeutic trials for its antibacterial and antioxidant (neuroprotective and anticancer) properties, as well as its other properties. Additional research is needed to extract and pinpoint the responsible bioactive elements in this plant. Successful clinical trials for illnesses that have not been sufficiently treated could lead to the creation of affordable but effective treatments that advance humanity.

## Figures and Tables

**Figure 1 plants-12-01354-f001:**
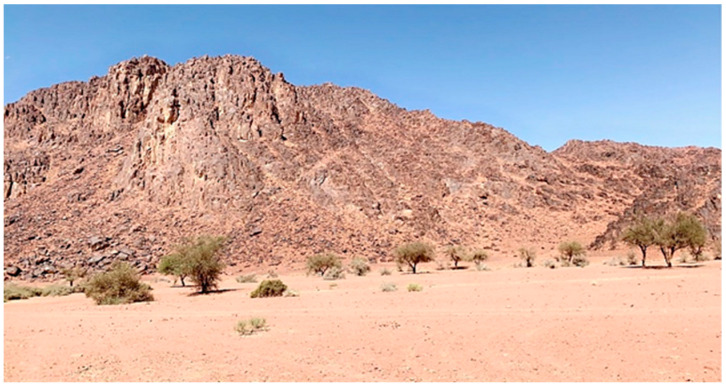
The Aja and Salma Mountains in Hail.

**Figure 2 plants-12-01354-f002:**
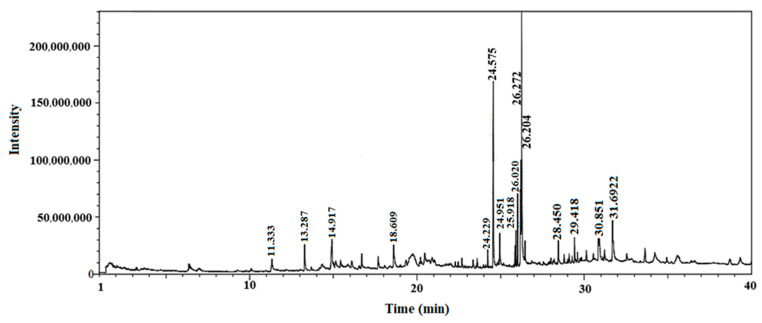
GC–MS chromatograms of *Fagonia indica* in methanolic extract (80% *v*/*v*).

**Figure 3 plants-12-01354-f003:**
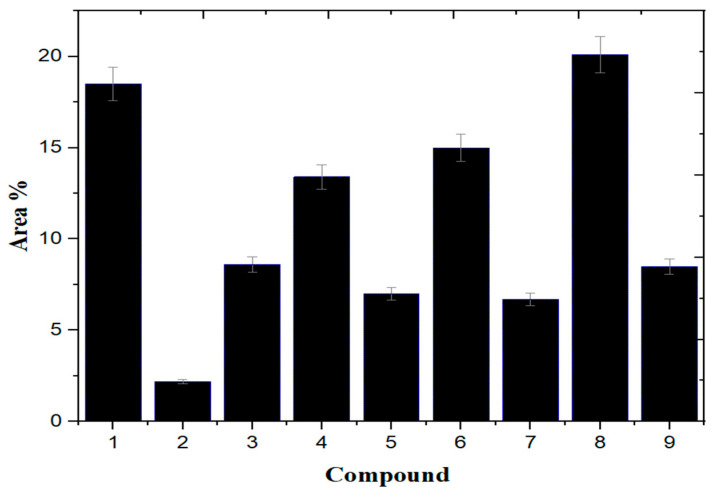
Percentages of chemical compounds extracted from *Fagonia indica* methanol extract.

**Figure 4 plants-12-01354-f004:**
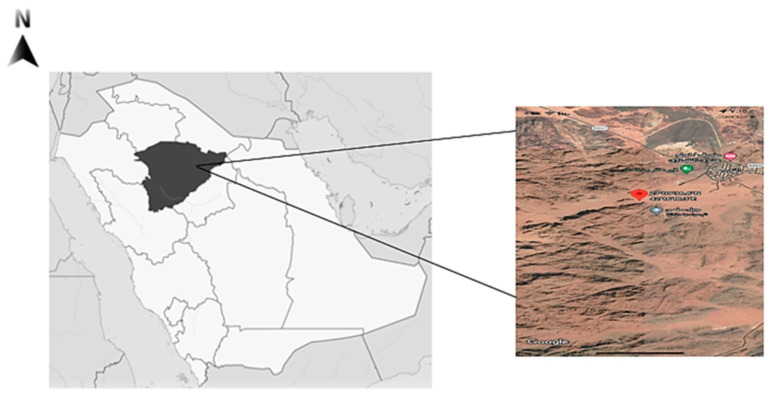
Map of Saudi Arabia showing the collection area of *Fogonia indica*.

**Figure 5 plants-12-01354-f005:**
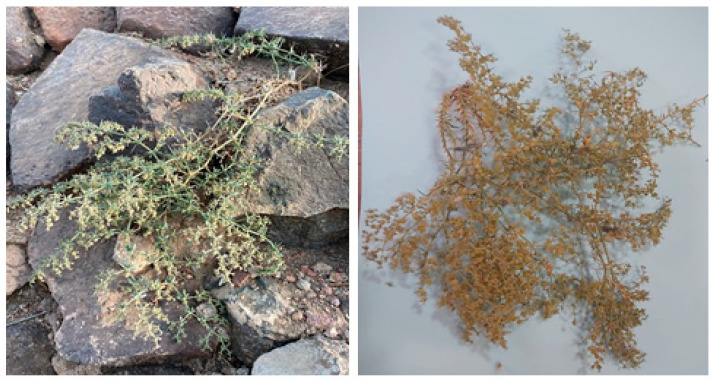
The areal parts of *Fagonia indica*.

**Table 1 plants-12-01354-t001:** Phytochemical profile of *Fagonia indica* methanol extract (80% *v*/*v*).

Plant Extract	Saponins	Terpenes	Flavonoids	Phenols	Tannins	Alkaloids (Dragendorf)	Alkaloids (Mayer)	Cardiac Glycosides
*Fogonia indica* Methanol extract 80% *v*/*v*	+	+	+	+	+	−	−	+

+: Present, −: Absent.

**Table 2 plants-12-01354-t002:** Chemical compounds in methanolic extract of *Fagonia indica* based on GC–MS.

No	Compound	Chemical Structure	Molecular Formula	R-Time	Area %
1	2-Chloropropanoic acid	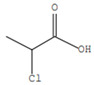	C_3_H_5_ClO_2_	1.4	18.5
2	Tridecanoic acid, 12-methyl-, methyl ester	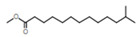	C_15_H_30_O_2_	22.9	2.2
3	Hexadecanoic acid, methyl ester	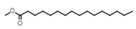	C_17_H_34_O_2_	24.2	8.6
4	Methyl 3-(3,5-di-tert-butyl-4-hydroxyphenyl)propionate	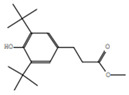	C_18_H_28_O_3_	24.3	13.4
5	Methyl linoleate	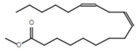	C_19_H_34_O_2_	25.9	7
6	Petroselinic acid methyl ester	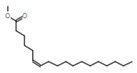	C_19_H_36_O_2_	25.9	15
7	Erucylamide	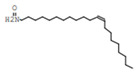	C_22_H_43_NO	31.7	6.7
8	Tetrahydro-2-methylfuran	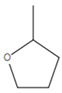	C_5_H_10_O	1.6	20.1
9	Diosgenin	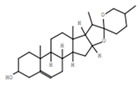	C_27_H_42_O_3_	1.4	8.5

**Table 3 plants-12-01354-t003:** Mineral composition of *Fagonia indica* (areal parts) using energy dispersive X-ray fluorescence analysis.

Compound	Concentration (Mass Percent, m/m%)	Std Err	Element	Concentration (Mass Percent, m/m%)	Std Err
CaO	48.28	0.25	Ca	34.52	0.18
SO_3_	17.96	0.19	S	7.19	0.08
K_2_O	13.37	0.17	K	11.10	0.14
Al_2_O_3_	6.68	0.15	AL	3.54	0.08
Cl	5.45	0.11	Cl	5.45	0.11
SiO_2_	3.10	0.13	Si	1.45	0.06
P_2_O_5_	2.03	0.18	P	0.884	0.08
F_2_O_3_	1.31	0.06	Fe	0.919	0.04
MgO	0.88	0.16	Mg	0.533	0.09
Na_2_O	0.53	0.15	Na	0.39	0.11
TiO_2_	0.166	0.008	Ti	0.099	0.005
SrO	0.110	0.005	Sr	0.093	0.005
ZnO	0.051	0.003	Zn	0.041	0.002
MnO	0.043	0.0041	Mn	0.033	0.003

**Table 4 plants-12-01354-t004:** Antioxidant activities of *Fagonia indica* methanol extract (80% *v*/*v*) as compared to known drugs.

Test System	Extract	Butylated Hydroxytoluene	Ascorbic Acid
Phytochemical compounds			
1. Total Phenols (mg GAE/g Extract)	3.72 ± 0.05	-	-
2. Total Tannins (mg TAE/g Extract)	24.14 ± 0.65	-	-
3. Total Flavonoids (mg QE/g Extract)	136.05 ± 0.45	-	-
Antioxidant Assays			
1. DPPH IC_50_ (mg/mL)	0.06 ± 0.003	0.023 ± 3 × 10^4^	0.022 ± 5 × 10^4^
2. ABTS IC_50_ (mg/mL)	0.07 ± 0.001	0.018 ± 4 × 10^4^	0.021 ± 0.001
3. β-carotene IC_50_ (mg/mL)	2.34 ± 0.04	0.042 ± 3.5 × 10^3^	0.017 ± 0.001

**Table 5 plants-12-01354-t005:** The inhibition zone of methanol extract of *Fagonia indica* expressed as means of three repetitions (mm + SD).

Bacteria Tested	1 mg/disc	2 mg/disc	3 mg/disc	Ampicillin 10 μg/disc
*Bacillus subtilis* (MTCC121)	12.0 ± 1.52	14.00 ± 1.5	15.00 ± 1.5	12.0 ± 1.0
*Pseudomonas aeruginosa* (MTCC741)	10.0 ± 1.0	11.0 ± 1.15	12.0 ± 1.0	6.0 ± 0.0

The zone of inhibition around discs impregnated with *Fagonia indica* methanol extract 80% *v*/*v* expressed as means of three repetitions (mm ± SD). SD: standard deviation.

**Table 6 plants-12-01354-t006:** Determination of MIC, MBC and MBC/MIC ratios of *Fogonia indica* methanol extract against the selected bacteria.

Microorganisms Tested	*Fogonia indica* Methanol Extract (80% *v*/*v*)		MBC/MIC
	MIC (μg/mL)	MBC (μg/mL)	
*B subtilis* (MTCC121)	125	250	4
*P. aeruginosa* (MTCC741).	250	500	16

**Table 7 plants-12-01354-t007:** Anti-biofilm formation of *Fagonia indica* against *B. subtilis* and *P. aeruginosa*.

** *B. subtilis* **		**Mean**			**SD**	**% Inhibition**
Control	0.62	0.65	0.69	0.65	0.034	
½ MIC	0.33	0.46	0.39	0.33	0.06	40.19
¼ MIC	0.42	0.52	0.59	0.51	0.08	21.60
** *P. aeruginosa* **		**Mean**			**SD**	**% Inhibition**
Control	0.85	0.85	0.88	0.86	0.01	
½ MIC	0.51	0.57	0.63	0.57	0.06	33.69
¼ MIC	0.68	0.72	0.73	0.70	0.03	17.78

## Data Availability

Data are contained within the article.

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
