# Peer review of "Unveiling Chemical, Antioxidant and Antibacterial Properties of Fagonia indica Grown in the Hail Mountains, Saudi Arabia"

_plants, 2023, doi:10.3390/plants12061354_

Round 1
Reviewer 1 Report
The peer-reviewed manuscript fits perfectly into the subject matter of the journal Plant, so I recommend it for publication. However, I would suggest some minor adjustments: 1. Latin names should be written in italics 2. The authors did not provide information (except for alkaloids) on what quality tests were performed (Tab.1) 3. The value "DPPH IC50 (mg/mL) 0.013 + 7.34" in Table 4 is probably an editorial error 4. There are no units in table 3 5. What type of grinder was used to grind the raw material? Using a grinder with a metal blade may falsify the results. 6. Are you sure the values have been correctly assigned in the sentence "Fagonia indica contained the following amounts of total phenols, total tannins, and total flavonoids: 3.72 + 0.05 mg GAE/g, 24.14 + 0.65 mg TAE/g, and 136.05 + 0.45 mg QE/g, respectively" I mean primarily the content of phenolic compounds.
Author Response
First, we would like to thank the Journal (Plants), the editor, and the reviewer for their valuable time and important comments. Our responses are in blue color
The peer-reviewed manuscript fits perfectly into the subject matter of the journal Plant, so I recommend it for publication. However, I would suggest some minor adjustments:
- Latin names should be written in italics. Thank you, all Latin names are now in font Italic.
- The authors did not provide information (except for alkaloids) on what quality tests were performed (Tab.1). Thank you, it is now explained in the “Materials and methods” section, see:
3.6. Phytochemical profile
The extracts were tested for fixed oils tannins, alkaloids, terpenoids, sterols, flavonoids, saponins, cardiac glycosides, fats, and phenols were carried out using colorimetric methods described elsewhere [54, 55].
- The value "DPPH IC50 (mg/mL) 0.013 + 7.34" in Table 4 is probably an editorial error. Yes, thank you for noticed that, the correct value for "DPPH IC50 (mg/mL) is 0.06 ± 0.003 (mentioned in the table and the text).
- There are no units in table 3. Thank you, the units were added (Mass percent, m/m %), and mentioned in the Table.
5.What type of grinder was used to grind the raw material? That is: (Electric Grinder,OMCG2145,Olsenmark, USA), we've updated the "Materials and Techniques" section accordingly.
Using a grinder with a metal blade may falsify the results. Thank you so much, the plant was ground dry not wet or fresh, we mentioned that the plant was dried in a shad for up to a week. many previous studies used such grinders:
https://doi.org/10.1016/j.tifs.2021.11.019
https://doi.org/10.1021/acsomega.1c03355
https://doi.org/10.1016/j.phyplu.2021.100167
- Are you sure the values have been correctly assigned in the sentence "Fagonia indica contained the following amounts of total phenols, total tannins, and total flavonoids: 3.72 + 0.05 mg GAE/g, 24.14 + 0.65 mg TAE/g, and 136.05 + 0.45 mg QE/g, respectively" I mean primarily the content of phenolic compounds. Thank you, The correct sentence: "Fagonia indica contained the following amounts of total phenols, total tannins, and total flavonoids: 3.73 ± 0.05mg GAE/g, 24.14 ± 0.6 mg TAE/g, and 136.05 ± 0.45 mg QE/g, respectively. “it is (±) not (+)”, this Inadvertent error was corrected in the text.

Reviewer 2 Report
An interesting, innovative article worth publishing.
The manuscript needs extensive editing and reformatting. Necessary changes in the methodological part.
All comments in the attached file.
For a major revision.

Author Response
We would like to thank the magazine, the editor, and the reviewer for their valuable time and important comments. Our responses are in blue color:
An interesting, innovative article worth publishing. Thank you so much, we appreciate that!
The manuscript needs extensive editing and reformatting. Necessary changes in the methodological part. Thank you so much, the manuscript has been extensively revised according to your valuable comment.
All comments in the attached file. Corrections were made according to the comments at the attached file and fixed in the last version of the manuscript ( pdf attached with this message) and the last version (revised) can be download from the Journal's page.
For a major revision.

Round 2
Reviewer 2 Report
As per previous comment, please provide FULL AND DETAILED descriptions in section 3.6 of the methodology.
Other issues have been corrected by the authors.
Author Response
Thank you so much, editor and reviewer, for the time you spent evaluating this manuscript. Our reply is in blue color:
As per previous comment, please provide FULL AND DETAILED descriptions in section 3.6 of the methodology. We appreciate your interest and scientific concern. We have abbreviated the methodology of section 3.6, since it is ordinary, but at your request, it has been included in more detail in the text.
Please find it tracked and highlighted in yellow color. Thank you so much.
Other issues have been corrected by the authors. We would like to express our sincere gratitude and appreciation for your feedback that contributed to the improvement.

Round 3
Reviewer 2 Report
Accept in present form.